# SENTENCE ENCODING WITH TREE-CONSTRAINED RELATION NETWORKS

## ABSTRACT

The meaning of a sentence is a function of the relations that hold between its words. We instantiate this relational view of semantics in a series of neural models based on variants of relation networks (RNs) which represent a set of objects (for us, words forming a sentence) in terms of representations of pairs of objects. We propose two extensions to the basic RN model for natural language. First, building on the intuition that not all word pairs are equally informative about the meaning of a sentence, we use constraints based on both supervised and unsupervised dependency syntax to control which relations influence the representation. Second, since higher-order relations are poorly captured by a sum of pairwise relations, we use a recurrent extension of RNs to propagate information so as to form representations of higher order relations. Experiments on sentence classification, sentence pair classification, and machine translation reveal that, while basic RNs are only modestly effective for sentence representation, recurrent RNs with latent syntax are a reliably powerful representational device.

## 1 INTRODUCTION

Sentence representations play a crucial role in many NLP tasks, including sentiment analysis, paraphrase detection, entailment recognition, summarisation, and machine translation. Popular approaches for encoding sentences compose words using recurrent units (Cho et al., 2014; Sutskever et al., 2014) or convolutions (Kalchbrenner et al., 2014; Kim, 2014). A series of papers has also explored improving sentence representations by incorporating structural information into the encoders, including both supervised syntactic information (Tai et al., 2015; Bowman et al., 2016; Ma et al., 2015; Socher et al., 2013), and unsupervised syntax where the structure is learned to maximise task performance (Yogatama et al., 2017; Choi et al., 2017).

We approach the problem of sentence representation starting from the insight that the meaning of a sentence is an aggregation of the set of relations that hold between words. For example, in the simple sentence "John saw Mary", most of the semantics is captured by knowing that "John is doing seeing" and that "Mary is what is being seen". Treating sentences as a conjunction of simple relational predicates has been influential in both formal semantics (Copestake et al., 2005) and roll-filler computational models of semantics (Smolensky, 1990). While the prior sentence representation architectures discussed above can implicitly discover and represent such relational facts, the recently proposed relation networks model (Santoro et al., 2017, RNs) lets us work with them more directly.

RNs represent a set of objects by aggregating representations of pairwise relations of all pairs of objects. For sentence representation, the "set" we seek to represent is a sentence, and the objects are words. There are two problems with this setup, and we develop extensions of RNs to address these shortcomings (§3). First, RNs assume equal contributions from all pairs of objects. In unstructured sets, this uniformity is justified, but sentences are structured sequences, not unstructured sets, and intuitively the relationship between some word pairs is more meaningful than the relationship between others. To address this structural deficiency, we extend RNs to be sensitive to pairwise attachment information based on a dependency tree structure. Our formulation lets us either use dependency trees obtained from external parsers, or learn trees as latent variables so as to optimise task performance. Tree learning is tractable in this model because we exploit the fact that RNs operate on representations of pairs of words, and we can efficiently obtain marginal probabilities of attachment decisions (i.e.,

word pairs) in pair-factored parsing models using Kirchoff's matrix tree theorem (Tutte, 1984; Koo et al., 2007; Liu & Lapata, 2017). A second objection to RNs is that they aggregate representations of pairs, but the meaning of a sentence may require understanding higher-order relational information (i.e., trees can be deeply nested). To enable representations that can directly capture the salient aspects of higher-order relations, we use recurrent RNs (Palm et al., 2017), but again augmented with supervised or latent dependency trees.

We carefully evaluate and analyse different versions of our RN-based models on machine translation and three text classification tasks, namely textual entailment, question type classification, and duplicate question detection (§4). Our experimental results demonstrate that in general sentence encoders based on recurrent RNs outperform those based on vanilla RNs, and models with latent tree constraints mostly achieve better performance than their supervised counter-parts. Overall, our models outperform sentence encoders based on birectional LSTMs and those incorporating an intra-sentence attention mechanism.

## 2 BACKGROUND

Relations networks (Santoro et al., 2017) are designed to model relational reasoning. Neural networks augmented with RNs have achieved success across various tasks such as visual question answering and text-based question answering on the bAbI suite of tasks, on which even models with sophisticated architecture could not obtain satisfactory results due to the requirement for complex relational inference. An RN is formalised as

$$\text{RN}(O) = f_\phi \left( \sum_{i,j} g_\theta(\mathbf{o}_i, \mathbf{o}_j) \right), \tag{1}$$

where the input is a set of "objects" $O = \{\mathbf{o}_1, \mathbf{o}_2, \ldots, \mathbf{o}_n\}$, $\mathbf{o}_i \in \mathbb{R}^d$, and $f$ and $g$ are functions parameterised by $\phi$ and $\theta$ using feed-forward neural networks (MLPs). The function $g_\theta$ computes the relatedness of two objects $\mathbf{o}_i$ and $\mathbf{o}_j$. RNs aggregate the relations between all the pairs of objects via the summation operation in Eq. 1. The embeddings of objects $\mathbf{o}_i$ and $\mathbf{o}_j$ differ depending on the task and the nature of the objects to be compared.

Although RNs perform well on problems that require basic relational reasoning, they are sub-optimal for tasks that demand multiple steps of reasoning as only pair-wise relational operations are involved. To address this limitation, Palm et al. (2017) proposed a recurrent variant for RNs, denoted as recurrent RNs, and demonstrated the model's ability for multi-step reasoning, which is essential for the tasks they studied, namely Sudoku and question answering. Recurrent RNs are formulated as follows:

$$\mathbf{m}_i^t = \sum_j g_\theta(\mathbf{h}_j^{t-1}, \mathbf{h}_i^{t-1}), \quad \mathbf{h}_i^t = f_\phi \left( \mathbf{h}_i^{t-1}, \mathbf{o}_i, \mathbf{m}_i^t \right). \tag{2}$$

At timestep $t$, $\mathbf{h}_i^t$ is the hidden state vector corresponding to the $i$-th object. In addition to the previous hidden state $\mathbf{h}_i^{t-1}$ and the input $\mathbf{o}_i$, it also takes into account the message $\mathbf{m}_i^t$, which summarises relations between the $i$-th object and all its neighbours (e.g. all objects in the simplest form) in the previous timestep $t - 1$. The number of recurrent steps is a hyperparameter, where higher values allow the modelling of longer chains of relations. In the standard setting, the vector $\mathbf{o}_i$ encodes timestep invariant information, such as position and ordering, and $g_\theta$ and $f_\phi$ are parameterised as MLPs and LSTMs (Hochreiter & Schmidhuber, 1997), respectively. In this architecture, the recurrent dependencies accumulate information of relations enabling multi-step reasoning.

## 3 MODELS

A straightforward application of RNs for sentence representation is to treat words as objects and consider all relations between pairs of words in a sentence following Eq. 1. Intuitively, this approach is unappealing as not all direct links between pairs of words are equally important. Thus, we propose to use a dependency grammar as a constraint in RNs and argue that the relations considered in the model should comply with a valid dependency tree structure. In the vanilla RNs, this means that the aggregations will operate on the subset of relations that form a valid tree (§3.1) and in the

recurrent RNs, we want the messages (Eq. 2) only coming from the syntactical neighbours of the node corresponding to a consistent tree structure across different timesteps (§3.2). In a supervised setting we use an off the shelf parser to generate trees for all sentences. In an unsupervised setting, we model them as latent variables.

## 3.1 RNs WITH TREE CONSTRAINTS

Let $\boldsymbol{x} = [x_1, x_2, \ldots, x_n]$ denote a sentence, where $x_i$ represents the $i$-th word in the sentence. Let $\boldsymbol{y}^*$ denote a directed dependency tree, which is represented by a set of head-modifier index tuples $(h, m)$.[1] To incorporate syntactic information, we modify Eq. 1 by restricting the summation function to span over $(h, m) \in \boldsymbol{y}^*$:

$$\mathbf{s} = f_\phi \left( \sum_{(h,m) \in \boldsymbol{y}^*} g_\theta(\mathbf{o}_h, \mathbf{o}_m) \right), \tag{3}$$

where $\mathbf{s}$ denotes the vector representation of the sentence $\boldsymbol{x}$, and $\mathbf{o}_h$ and $\mathbf{o}_m$ are embeddings corresponding to the head word $x_h$ and the modifier word $x_m$, respectively.

To enable task-optimal trees to be learned we introduce a distribution over trees given the sentence, $p_\psi(\boldsymbol{y} \mid \boldsymbol{x})$, and rewrite Eq. 3 in terms of expectations under this distribution:

$$\mathbf{s} = f_\phi \left( \sum_{\boldsymbol{y} \in \mathcal{Y}(\boldsymbol{x})} p_\psi(\boldsymbol{y} \mid \boldsymbol{x}) \sum_{(h,m) \in \boldsymbol{y}} g_\theta(\mathbf{o}_h, \mathbf{o}_m) \right),$$

where $\mathcal{Y}(\boldsymbol{x})$ denotes the set of dependency trees for a sentence $\boldsymbol{x}$. Because the inner loop depends only on $(h, m)$ pairs and the linearity of expectation, we can rewrite this in terms of the marginal edge probabilities $p_\psi(h \rightarrow m \mid \boldsymbol{x}) = \sum_{\boldsymbol{y} \in \mathcal{Y}(\boldsymbol{x})} p_\psi(\boldsymbol{y} \mid \boldsymbol{x})[(h, m) \in \boldsymbol{y}]$, where $[q] = 1$ if $q$ is true and 0 otherwise. This becomes

$$\mathbf{s} = f_\phi \left( \sum_{(h,m) \in \mathcal{D}(\boldsymbol{x})} p_\psi(h \rightarrow m \mid \boldsymbol{x}) g_\theta(\mathbf{o}_h, \mathbf{o}_m) \right),$$

where $\mathcal{D}(\boldsymbol{x})$ denote all possible dependencies arcs for a sentence $\boldsymbol{x}$: $\mathcal{D}(\boldsymbol{x}) = \{(h, m) \mid h \in [0 \ldots n], m \in [1 \ldots n]\}$.

Fortunately, the required attachment marginals can be calculated efficiently in $O(n^3)$ using Kirchhoff's matrix-tree theorem (Tutte, 1984), provided the conditional tree probability is a conditional random field with potentials that only look at a single pair of nodes at a time (Koo et al., 2007; Smith & Smith, 2007; Liu & Lapata, 2017). Although this restriction on the potential functions may seem limiting (the model cannot look at higher order dependency structures when assigning probability to the tree), representations derived from neural networks employing such models can produce state-of-the-art supervised dependency parsers (Dozat & Manning, 2017), suggesting this is an acceptable restriction.

For our model, we define the potential associated with an edge using a bilinear form (Ma & Hovy, 2017),

$$\Psi(h, m) = e^{\mathbf{o}_h^T \mathbf{W} \mathbf{o}_m + \mathbf{U}^T \mathbf{o}_h + \mathbf{V}^T \mathbf{o}_m + b}, \tag{4}$$

where $\mathbf{o}_i$ denotes the BiLSTM output corresponding to the input word $x_i$; $\mathbf{W}$, $\mathbf{U}$, $\mathbf{V}$ are weight matrices, and $b$ is the bias term.

**Structured intra-sentence attention**  For many applications, it is useful to represent each word in the sentence in terms of its context in the surrounding sentence. While simple mechanisms like bidirectional RNNs can produce satisfactory results (Bahdanau et al., 2015), attention-based word-in-context representations have received increasing interest (Vaswani et al., 2017), including

---

[1]For the experiments below that use syntax trees, we obtain dependency parses from SyntexNet (Weiss et al., 2015; Kong et al., 2017), an open-source parser trained on the Universal Dependencies English dataset (Nivre et al., 2016).

those that impose structural constraints on attention (Liu & Lapata, 2017; Kim et al., 2017). Our tree-constrained RN can likewise be used to construct word-in-context representations. Intuitively, we create a contextualised representation for each word by aggregating information from its possible parents and children according to their attachment marginal probability. These context vectors and word vectors can either be operated on by models that require them or pooled over to generate a sentence representation.

For RNs with supervised tree constraints, we define the context vector for the word $x_i$ in terms of the representation of the relation between $x_i$ and its parent $x_h$, i.e., $\mathbf{r}_i = f_\phi(g_\theta(\mathbf{o}_h, \mathbf{o}_i))$.

For latent tree RNs, the context vector for each word $x_i$ contains two components: a weighted sum of relations between $x_i$ and its parents, and a weighted sum of relations between $x_i$ and its children:

$$\mathbf{p}_i = f_\phi\left(\sum_{h=1}^{n} p_\psi(h \to i | \boldsymbol{x}) g_\theta(\mathbf{o}_h, \mathbf{o}_i)\right), \mathbf{c}_i = f_\phi\left(\sum_{m=1}^{n} p_\psi(i \to m | \boldsymbol{x}) g_\theta(\mathbf{o}_i, \mathbf{o}_m)\right), \mathbf{r}_i = [\mathbf{p}_i; \mathbf{c}_i].$$

Finally, the $\mathbf{r}_i$'s are concatenated with the original object representations and transformed into the final contextual word representations: $\mathbf{s}_i = \tanh(\mathbf{W}_r^T[\mathbf{r}_i \; ; \; \mathbf{o}_i])$.

For applications where a single vector is required, a subsequent pooling operation is applied to the sequence of vectors $[\mathbf{s}_1, \mathbf{s}_2, \dots, \mathbf{s}_n]$ to generate an alternative representation of a sentence $\mathbf{s}$. We discuss the pooling operations used below (§4 and §A.1).

In our model, the context vectors are computed from the representations of word pairs (i.e., components of RNs), whereas in the related prior work of Liu & Lapata (2017) and Kim et al. (2017), they are the expected value of LSTM hidden state vectors under structured attention distributions.

## 3.2 RECURRENT RNS WITH TREE CONSTRAINTS

The models described in the previous sections are based on vanilla RNs which aggregate only single pairs of words. Since language can have deeply nested structures, even if an RN can faithfully represent the contents of sentence, the representation may not represent important higher order dependencies in a convenient form. To deal with this limitation, we augment *recurrent* RNs with tree constraints. In this model, information is propagated along the edges in the (possibly latent) tree structure associated with each sentence during the construction of the representation.

We provide two variants of recurrent RNs, one for supervised trees and the other for latent trees. With supervised trees, the message of an object at each timestep comes only from its parent in the given dependency parse. Formally, let $\boldsymbol{y}^*$ denote the given parse tree,[2]

$$\mathbf{m}_i^t = \sum_{(h,i) \in \boldsymbol{y}^*} g_\theta(\mathbf{h}_h^{t-1}, \mathbf{h}_i^{t-1}), \quad \mathbf{h}_i^t = f_\phi(\mathbf{h}_i^{t-1}, [\mathbf{o}_i \; ; \; \mathbf{m}_i^t]),$$

In the unsupervised setting, where the tree structure is latent, the messages from neighbours could include those from all the possible parents or children or both.[3] We include both messages here:

$$\mathbf{m}_i^{t,p} = \sum_{h=1}^{n} p_\psi(h \to i \mid \boldsymbol{x}) g_\theta(\mathbf{h}_h^{t-1}, \mathbf{h}_i^{t-1}), \quad \mathbf{m}_i^{t,c} = \sum_{m=1}^{n} p_\psi(i \to m \mid \boldsymbol{x}) g_\theta(\mathbf{h}_i^{t-1}, \mathbf{h}_m^{t-1}),$$
$$\mathbf{h}_i^t = f_\phi(\mathbf{h}_i^{t-1}, [\mathbf{o}_i \; ; \; \mathbf{m}_i^{t,p} \; ; \; \mathbf{m}_i^{t,c}]).$$

The tree probabilities remain fixed across the different timesteps of the recurrent RN, ensuring that multi-step reasoning operates on a consistent tree structure. The initial hidden representation of each object $\mathbf{h}_i^1$ is set to $\mathbf{o}_i$. Finally, we pool over the hidden state vectors of the last timestep $T$ for sentence representations, or make them available as word-in-context representations.

---

[2]There will be only a single parent for each $i$ in $\boldsymbol{y}^*$, but we use this notation for parallelism with the model below.

[3]Empirically, we find that messages from parents are crucial, but adding extra messages from children gives a very small improvement.

## 4 EXPERIMENTS

We examine the effectiveness of our sentence representation models on three sentence classification tasks: textual entailment, question type classification, and paraphrase detection. We further use attention-based machine translation to test our proposed methods' word-in-context representations. We refer the readers to §A.1 for implementation details.

### 4.1 SENTENCE CLASSIFICATION

In our experiments, we use the Stanford Natural Language Inference (SNLI), Quora duplicate question detection, and TREC question type classification datasets.

**Stanford natural language inference**   Given a premise and hypothesis pair $(\boldsymbol{x}_1, \boldsymbol{x}_2)$, the task in SNLI is to predict whether their relationship is *entailment*, *contradiction*, or *neutral*. The dataset consists of 550k, 10k, and 10k sentences for training, development, and test set, respectively (Bowman et al., 2015).

**Quora duplicate question detection**   The dataset[4] contains sentence pairs mined from questions that people asked on Quora. The task is to identify whether a question is a duplicate of another. We follow the data splits by Wang et al. (2017), having 400k/10k/10k for train/dev/test dataset.

**TREC question type classification**   The task (Li & Roth, 2002) is to classify open-domain, factoid questions into six categories: abbreviation, entity, description, human, location, and numeric value. The dataset is composed of 5.5k examples in the training dataset and 500 examples in the test set. We randomly sample 500 examples from the training set for validation and leave the rest for training.

**Classification model**   In these classification tasks, we calculate the probability distribution of labels as follows: $p(Y \mid \mathbf{q}) = \mathrm{softmax}(\mathbf{W}_q^T \boldsymbol{\phi}(\mathbf{q}) + \mathbf{b}_q)$, where the function $\phi$ is an MLP with 1 hidden layer. For question type classification, the vector $\mathbf{q}$ is simply the sentence representation $\mathbf{s}$. For the other tasks which classify pairs of sentences $(\boldsymbol{x}_1, \boldsymbol{x}_2)$, $\mathbf{q}$ is obtained by concatenating the following vectors: the representations of the two input sentences $\mathbf{s}^{\boldsymbol{x}_1}$ and $\mathbf{s}^{\boldsymbol{x}_2}$, element-wise product $\mathbf{s}^{\boldsymbol{x}_1} \odot \mathbf{s}^{\boldsymbol{x}_2}$, and absolute element-wise difference $|\mathbf{s}^{\boldsymbol{x}_1} - \mathbf{s}^{\boldsymbol{x}_2}|$ (Tai et al., 2015; Bowman et al., 2016).

**Classification results**   Tables 1a, 1b, and 2a summarise the results of our RN-based models for the three text classification tasks. We also include three baselines, namely the bag-of-words model (BoW), the BiLSTM encoder with max[5] pooling layer, and a re-implementation of the structured intra-sentence attention model of Liu & Lapata (2017).[6] The models with an intra-sentence attention mechanism perform on par or better than the vanilla RNs; and on average those based on recurrent RNs achieve the best results across all datasets. Tree constraints add little value to vanilla RNs. Given that the underlying RN formulation fails, it is unsurprising that tree constraints do not add value (since adding tree constraints is simply adding a probability distribution to the underlying RN architecture). RNs become more effective when we address the bottleneck of simple aggregation of relations by introducing the intra-sentence attention mechanism. In this case, tree constraints reliably begin to work. Finally, we find that models with latent trees outperform those with supervised trees, supporting the patterns of results found across a range of models explored in prior work (Williams et al., 2017; Yogatama et al., 2017; Choi et al., 2017).

For SNLI, compared to the published results from other sentence encoding-based models,[7] our best model achieve slightly better results than the model by Choi et al. (2017), which encodes sentences

---

[4]https://data.quora.com/First-Quora-Dataset-Release-Question-Pairs

[5]Max was empirically determined to be optimal by us as well as in previous work (Conneau et al., 2017).

[6]Since we are interested in comparing models based on individual words to models based on relations, we make two small changes to the Liu & Lapata (2017) model to make it maximally comparable to our RN model. First, we use the $\mathbf{o}_i$'s both for calculating attention and the summed representation (the prior work splits these into two parts); we also use the vector-difference representation to make the classification decision, rather than a sentence-matching module.

[7]For a complete list of results on this task, we refer the readers to the leaderboard https://nlp.stanford.edu/projects/snli/.

Table 1: (a). Results on SNLI. The LSTM encoder and SPIINN-PI refer to Bowman et al. (2016). The results of BiLSTM with max-pooling, Gumbel TreeLSTM encoder, reinforced self-attention are obtained from Conneau et al. (2017); Choi et al. (2017) and Shen et al. (2018), respectively. (b). Results on Quora duplicate questions detection. BiMPM and pt-DECATT (*) incorporate complex mechanism for mapping sentences; and the remainder of the models are sentence-encoding models.

| System | Acc. |
|---|---|
| **Published models** | |
| 300D LSTM encoder | 80.6 |
| 300D SPINN-PI encoder | 83.2 |
| 4096D BiLSTM with max-pooling | 84.5 |
| 300D Gumbel TreeLSTM encoder | 85.6 |
| 300D Reinforced self-attention | **86.3** |
| **Baselines** | |
| BoW | 77.9 |
| BiLSTM with max pooling | 85.1 |
| Structured attention | 84.5 |
| **Our models** | |
| RNs + no tree | 84.1 |
| RNs + supervised tree | 83.7 |
| RNs + latent tree | 83.6 |
| RNs intra-attn + no tree | 84.0 |
| RNs intra-attn + supervised tree | 84.7 |
| RNs intra-attn + latent tree | 85.2 |
| Recurrent RNs + supervised tree | 85.2 |
| Recurrent RNs + latent tree | **85.7** |

(a)

| System | Acc. |
|---|---|
| **Published models** | |
| CNN (Wang et al., 2017) | 79.6 |
| LSTM (Wang et al., 2017) | 82.6 |
| GRU (Nicosia & Moschitti, 2017) | 86.8 |
| *BiMPM (Wang et al., 2017) | 88.2 |
| *pt-DECATT (Tomar et al., 2017) | **88.4** |
| **Baselines** | |
| BoW | 81.3 |
| BiLSTM with max pooling | 86.8 |
| Structured attention | 86.8 |
| **Our models** | |
| RNs + no tree | 86.4 |
| RNs + supervised tree | 86.7 |
| RNs + latent tree | 87.0 |
| RNs intra-attn + no tree | 85.9 |
| RNs intra-attn + supervised tree | 87.2 |
| RNs intra-attn + latent tree | **87.4** |
| Recurrent RNs + supervised tree | 87.1 |
| Recurrent RNs + latent tree | 87.0 |

(b)

with a TreeLSTM (Tai et al., 2015) and learns a latent tree using the Gumbel-softmax (Jang et al., 2016). Our model is slightly outperformed by the very recent model of Shen et al. (2018) that encode sentences using a hybrid of hard and soft attention mechanisms.

On Quora duplicate question detection, our models outperform the CNN and LSTM baselines presented in previous work by a large margin. The results produced by our best model is close to BiMPM (Wang et al., 2017) and pt-DECATT (Tomar et al., 2017), which incorporate complex inter-sentence attention during sentence encoding.

## 4.2 MACHINE TRANSLATION

To further evaluate the effectiveness of RNs on text processing, we apply our models to machine translation. In the standard seq2seq model with attention (Bahdanau et al., 2015), the output of the encoder is a sequence of annotation vectors, which is the concatenation of the output of a BiLSTM at each timestep. These vectors are then used to derive the context vectors that are used by the decoder to predict the output tokens. We adapt the standard attentive seq2seq model by replacing the annotation vectors with the sequence of vectors $[\mathbf{s}_1, \mathbf{s}_2, \ldots, \mathbf{s}_n]$, comparing the intra-sentence attention (§3.1) or the recurrent RN (§3.2) definitions of these.

We experiment on Chinese–English and English–Vietnamese machine translation tasks here in order to evaluate our models' capability of capturing syntactic information of sentences. For Chinese–English, we use parallel data with 184k sentence pairs from the FBIS corpus (LDC2003E14). The training set is preprocessed in the same way as that in Yu et al. (2017). For English–Vietnamese, the dataset that we evaluate on is IWSLT 2015, containing 133k training sentence pairs obtained from scripts of TED talks. We follow prior work on dataset split and data preprocessing (Luong & Manning, 2015; Huang et al., 2018).

Table 2b shows the results of our models in comparison with standard attention based benchmarks (Bahdanau et al., 2015) and published models. While in classification tasks, recurrent RNs and latent

Table 2: (a) Results on TREC question type classification. (b) BLEU scores of different models on Chinese→English and English→Vietnamese machine translation. The results of neural noisy channel, Standford NMT, and neural phrase-based MT are from Yu et al. (2017); Luong & Manning (2015); Huang et al. (2018), respectively. We report the average of 5 results for the baselines and our models; * indicates the difference between this score and the score from the standard attention baseline is significant according to an approximate randomisation test (Clark et al., 2011).

| System | Acc. |
|---|---|
| **Published models** | |
| CNN (Kim, 2014) | 93.6 |
| SVM (Kim, 2014) | **95.0** |
| **Baselines** | |
| BoW | 85.1 |
| BiLSTM with max pooling | 93.1 |
| Structured attention | 91.7 |
| **Our models** | |
| RN + no tree | 93.1 |
| RN + supervised tree | 92.5 |
| RN + latent tree | 92.9 |
| RN intra-attn + no tree | 92.7 |
| RN intra-attn + supervised tree | 92.3 |
| RN intra-attn + latent tree | 92.1 |
| Recurrent RNs + supervised tree | 94.0 |
| Recurrent RNs + latent tree | **94.2** |

(a)

| System | BLEU zh-en | BLEU en-vi |
|---|---|---|
| **Published models** | | |
| Neural noisy channel | 26.4 | – |
| Stanford NMT | – | 23.3 |
| Neural Phrase-based MT | – | **27.7** |
| **Baselines** | | |
| Standard attention | 26.5 | 26.2 |
| Structured attention | 26.4 | 26.0 |
| **Our Models** | | |
| RNs intra-attn supervised tree | 26.7 | 26.4 |
| RNs intra-attn latent tree | *27.1 | *26.5 |
| Recurrent RNs supervised tree | 26.1 | 26.2 |
| Recurrent RNs latent tree | 26.4 | *26.5 |

(b)

(a) Learned tree on TREC

(b) Learned tree on SNLI

(c) Learned tree on SNLI

(d) Tree given by a parser

(e) Learned tree on TREC

(f) Tree given by a parser

Figure 1: Parse trees induced by our RN-based models and trees given by the dependency parser. The discussion of these trees is presented in §5.

trees work best, in translation, flat RNs with latent trees work best. We suggest that this is due to the nature of the translation task, which can be accomplished by having a good representation of each word in context.

# 5 ANALYSIS

To understand the behavior of different RN-based models, we select a few examples to examine different types of errors they made, and inspect the learned latent dependency parse trees. Figure 1 shows the parse trees learned by RN-based models with latent tree constraints. These trees are maximum spanning trees obtained from Chu-Liu-Edmonds algorithm (Chu & Liu, 1965; Edmonds, 1967) over the scores of word pairs (Eq. 4). We observe that the induced trees are optimised for

different tasks and generally are not as linguistically intuitive as those produced by a dependency parser trained in a supervised fashion. For question type classification, the type of a question can usually be decided by one or two words. Therefore the learned trees are intended to have the key words as roots that are depended on by all the other words. For example, in Fig. 1a, the word *where* completely determines the type of the question *Where is the Orinoco River?* to be *location* and it serves as the root of the learned tree. For SNLI, the dependencies in the learned trees (Fig. 1b) are less focused on a few words, probably due to the fact that a precise judgment for entailment requires the accumulation of information over the entire sentence.

The learned trees, although not necessarily linguistically plausible, tend to have direct connections between words whose relations are crucial for the down-stream task. As an example, consider the sentence pair from SNLI:

*A man uses a projector to give a presentation.    A man is using a projector to watch television.*

Fig. 1c shows the tree of the premise induced by the model 'RNs intra-attn + latent tree'. The word *presentation* is directly linked to the word *projector*, and this dependency helps the model to realise that the projector is used for a presentation rather than watching TV, and therefore predict their relationship as *contradiction*. By contrast, as shown in Fig. 1d, there are multiple hops between these two words in the parse tree given by an external parser. The model 'RNs intra-attn + supervised tree', which is based on this parse, fails to yield the correct answer.

In contrast, Fig. 1e and 1f illustrate an example where the models with latent trees failed but supervised trees were successful. In this case, the latent model has drawn attention to *what*, which is generally a good strategy for this task, but in this case the restrictor *birthday* carries the crucial evidence that the answer is *number*.

Comparing the output of vanilla RNs and recurrent RNs (both with tree constraints), we find that recurrent RNs are better at capturing multi-step inference. For instance, to decide the type of the question *What is the statue of liberty made of?* the model should capture the relations between *What* and *statue of liberty*, *statue of liberty* and *made*, *made* and *of*. While recurrent RNs classify its type correctly (as *entity*), vanilla RNs misclassify it as *description*, which would be the type of the prefix *What is the statue of liberty*.

## 6 RELATED WORK

In sentence encoders, standard neural-network architectures for composing word embeddings are convolutional neural networks (CNNs) (Kalchbrenner et al., 2014; Kim, 2014), and LSTMs (Conneau et al., 2017). However, there has been considerable effort in embedding tree structure to modulate the behavior of encoders (Bowman et al., 2016; Tai et al., 2015; Socher et al., 2013). More recently, a series of work has developed methods for learning custom tree structures to optimise downstream performance (Yogatama et al., 2017; Maillard et al., 2017; Choi et al., 2017).

The intra-sentence attention mechanism we used in our model was introduced by Cheng et al. (2016); Parikh et al. (2016). Subsequently, Liu & Lapata (2017) and Kim et al. (2017) added structural biases to the attention, making them induce latent dependency trees.

Our model encodes sentences by explicitly computing the relations between words in a sentence. Another type of model that involves relation-centric computation is graph convolutional networks (Kipf & Welling, 2017), which have been applied to the tasks of semantic role labelling (Marcheggiani & Titov, 2017), machine translation (Bastings et al., 2017), and knowledge base completion (Schlichtkrull et al., 2017).

## 7 CONCLUSION

In this paper we developed a new strategy for encoding sentences based on two extensions to relation networks (Santoro et al., 2017; Palm et al., 2017) that help them represent structured sentences by using dependency tree information, and that help them represent higher-order relations using an iterative message passing approach to computing representations. These enhancements significantly improve the quality of relation-based representation across a variety of tasks, and make them comparable to best existing encoders.

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

# A APPENDIX

## A.1 IMPLEMENTATION DETAILS

For all the experiments, hyperparameters are optimised via grid search on the accuracy or perplexity of the validation set.

In classification tasks, the models are optimised either with Adam (Kingma & Ba, 2014) with learning rate 0.0001 or Adagrad (Duchi et al., 2011) with learning rate 0.01, as determined by the performance on the validation set. Word embeddings are initialised with 300 dimensional GloVe vectors trained on Common Crawl 840B[8] (Pennington et al., 2014) and finetuned during training. The output of BiLSTM is fed as input to the RNs. The number of hidden units for BiLSTMs is 300. For the functions $g_\theta$ and $f_\phi$ in RNs, we use two-hidden layer MLPs, consisting of 300 hidden units per layer with ReLU non-linearities. We add residual connections to the MLPs that parameterise $g_\theta$ and $f_\phi$. Dropout (rate 0.5) is applied to the input and output of BiLSTMs. We replace the sum operations in Eq. 1 by max and use max pooling in the final layer of our model with intra-attention and the model based on recurrent RNs.

In both machine translation tasks, the functions $f_\phi$ and $g_\theta$ are parameterised using two-hidden layer MLPs, consisting of 512 hidden units per layer with ReLU. A beam size of 10 is used for decoding. For English-Vietnamese, we use 1 layer BiLSTM with 512 hidden units as encoder and 2-layer unidirectional LSTM as decoder. The model is optimised using stochastic gradient descent with learning rate 0.2 and dropout 0.5. For Chinese-English, the encoder is 1 layer BiLSTM and the decoder is 1 layer unidirectional LSTM (512 hidden units). We use Adam (Kingma & Ba, 2014) as optimiser with learning rate 0.0001. Dropout rate is 0.5. For all experiments with recurrent RNs, we set the number of recurrent steps to 3.

---

[8]https://nlp.stanford.edu/projects/glove/

