# OpenReview forum: "Sentence Encoding with Tree-Constrained Relation Networks"
_ICLR.cc/2019/Conference_

### Official Review · AnonReviewer3 · 2018-11-02
**This paper presents a competitive baseline method for generic sentence representation learning.**

**Rating:** 5
**Confidence:** 4

**Review:**

The main idea is to incorporate linguistic-based constrains in the form of dependency trees into different variations of relation networks.
In general, the paper is well written and organized, the presented problem is well motivated, and the approach is very strait forward. The experimental setting is comprehensive, and the results are indeed competitive in a wide range of tasks.
I think that using linguistic knowledge to improve Neural networks performance is very promising field, I think that you could get a much more substantial gains when applying your method in less resource-rich setups (maybe using some small subset of training for the SNLI and question duplication datasets).
It seems that your method relies heavily on previous works (RN, RNN-RN, latent dependency trees ,intra-sentence attention), can you please state clearly what your contribution is? does your model has any advantages over current state-of-the-art methods?

edit: I'm still not convinced about this article novelty, I really like the overall idea but it seems that this kind of contribution is better suited for short paper.

---

> ### Author Response · Authors · 2018-11-26
> **Author response**
>
> We thank the reviewer's comment. Please refer to the general response for the novelty of the paper.

---

### Official Review · AnonReviewer2 · 2018-11-03
**The actual effectiveness of the proposed method is unclear**

**Rating:** 5
**Confidence:** 4

**Review:**


[Summary]
The main purpose of this paper is to propose an extension of relation networks.
The proposal consists of two parts: 1) to integrate constraints of dependency syntax to control which relations influence the representation, and 2) utilize a recurrent computation to capture higher-order relations.

[clarity]
This paper is basically well written.
Motivation and goal are clear.

[originality]
The idea of utilizing supervised or unsupervised dependency tree as constraints to control which relations influence the representation seems novel and interesting.
However, technically it consists of the combination of the previous methods, such as matrix-tree theorem for calculating conditional probabilities, and structured attention.
Therefore, the proposed method is incremental rather than innovative.

[significance]
Experiments on several varieties of datasets revealed that the proposed method consistently improved the performance from the baseline RN.
In contrast, it did not outperform the current best scores for all experiments comparing with the current published best methods.
Obviously, we have no reason that we must use RNs for such tasks.
Therefore, the actual effectiveness of the proposed method in terms of the actual task settings is unclear for me.
I concern about the actual calculation speed of the proposed method.
The proposed method seems to require much higher computational cost against the baseline RNs.

[Questions]
1, Regarding the approach in general, it would be nice to see how much it depends on the quality of the dependency parse.
For example, we cannot always prepare a good parser for experiments on MT such as low-resource languages.
Do you have any comments for this?

2, Some experimental results showed that “RN intra-attn” was better than “Reccurent RNs”.
This implies for me that the higher-order dependency is useless for such tasks.
Are there any analyses why “Reccurent RNs” did not work well?

---

> ### Author Response · Authors · 2018-11-26
> **Author response**
>
> We proposed two versions of tree-constraints to RNs. While the supervised tree constraint may be affected by the quality of dependency parsers, the unsupervised tree constraint cannot be affected. In general, model with learnt trees perform better than the trees given by the dependency parsers.
>
> Recurrent RNs perform better than RNs+self-att on classification tasks and slightly worse on MT. We did not do in-depth analysis of why recurrent RNs are doing worse on MT, but our intuition is that MT arguably requires more surface form matching than ``interpretation'' of the sentence.

---

### Official Review · AnonReviewer1 · 2018-11-03
**motivation is confusing, needs to be better situated relative to related work**

**Rating:** 3
**Confidence:** 4

**Review:**

The paper presents an extension of relation networks (RNs) for natural language processing. RNs are designed to represent a set as a function of the representations of their elements. This paper treats a sentence as a set of words. Whereas regular RNs assume that the representation of a set is a uniform aggregation of the representation of the pairs of elements in the set, this paper proposes to weight the relevance of the pairs according to their tree-structured dependency relations between the words. The authors evaluate on a suite of NLP tasks, including SNLI, Quora duplicate question ID, and machine translation. They show marginal improvements over naive baselines, and no improvement over SOTA.

I am concerned about both the motivation for and the novelty of this work. My reading of this work is that the authors try to reverse engineer a TreeRNN in terms of RNs, but I am not sure what the reason is for wanting to use the RN framework in order to derive an architecture that, IIUC, essentially already exists. I can't find any fundamentally meaningful differences between the proposed architecture and the existing work on TreeRNNs, and the results suggest that there is nothing groundbreaking being proposed here. It is possible I am missing some key insight, but I do believe the burden is on the authors to highlight where the novelty is. The intro *and* related work sections should both be rewritten to answer the question: what is the insufficiency with current sentence encoding models that is addressed by this architecture? Currently, the intro addresses the tangential question: what is the insufficiency with RNs for NLP that is addressed by this architecture? If the latter is the question the authors want to answer, they need to first answer: why should we want to cast sentence encoders as RNs as opposed to any of the (many) other available architectures? Without a firmer understanding of what this paper contributes and why, I can't recommend acceptance. More detailed comments for the authors below.

- You introduce a few naive baselines, but none of these is a TreeRNN. TreeRNNs are the obvious baseline, and you should be comparing on each and every evaluation task, even if there is no previously published result for using tree RNNs on that task. For the one result (SNLI, table 1) on which there is previous work using TreeRNNs, the table confirms my intuition that the proposed model is no improvement over the TreeRNN architecture. It seems very important to address this comparison across all of the evaluation tasks.
- I like the notion of marginalizing over latent tree structures, but the related work section needs to make clear what is being contributed here that is different from the cited past work on this problem
- On the MT eval, why are you missing values for zh-en on the NMT models that are actually competitive? I think many of these models are open-source or easy to reimplement? Its hard to draw conclusions when from such a gappy table.
- Only table 2 has significance values (over naive baseline that is) which implies that the other results are not significant? That is disconcerting.
- I am disappointed in the analysis section. As is, you provide an ad-hoc inspection of some inferred trees. I find this odd since there is no evidence that the tree-ness of the architecture (as opposed to, e.g., recurrence or attention) is what leads to quantitative improvements (at least according to the experimental results in the tables), so there is no reason we should actually expect the trees to be good or interesting. My interpretation of these cherry-picked examples is that the learning is fighting the architecture a bit, basically "learning a tree" that reduces to being an attention mechanism that up-weights one or two salient words.
- The analysis I *wanted* to see instead was why recursion helped for sentence classification, it did not for MT. You give an intuition for this result but no evidence. (That is assuming that, quantitatively, this trend actually holds. Which maybe is not the case if none of the results are significant.)
- In general, regarding evaluation, SNLI is overfit. You should use MNLI at least. I have trouble interpreting progress on SNLI as "actual" progress on language representation.
- The related work section as a whole is too short. If you need to cut space, move technical content to appendix, but don't compromise in related work. You listed many relevant citations, but you have no context to situate your contribution relative to this past work. What is the same/different about your method? You should provide an answer to that for each and every paper you cite.

---

> ### Author Response · Authors · 2018-11-26
> **Author response**
>
> We would encourage the reviewer to be less aggressive even when writing a negative review. We put a lot of effort into this paper, and we wrote it honestly and in good faith. We respond to the substantive criticisms in what follows:
>
> - Adding TreeRNN as a baseline
> We do not think it is necessary to add TreeRNN as a baseline because the main goal of the paper is to explore whether relation-based sentence representation is effective. We have compared our model with the strong BiLSTM baseline (in fact we achieved better results on BiLSTM than what have been reported). For each task, we also compare our model with existing state-of-the-art models, which are generally word-based models.
>
> - Missing baselines for zh-en in MT
> This is a dataset that has been used previously in the literature (indeed, it has been widely used since even before NMT), and is a good baseline to replicate since it is not inconveniently large, which is an important standard to have to encourage research on interesting models. We additionally chose it because Chinese and English have a relatively large amount of word order differences, and have previously benefited from syntax-based translation. We are not intentionally missing values on competitive models, but the values that we put in the table are those we can find in existing papers.
>
> - Significance values
> We focused on reporting significance on MT since the pattern of improvements was different for MT than for the other tasks (which told a reliable story). Since we are attempting to saying something about the model in general (and not individual experimental differences), we did not see the value of reporting standard hypothesis tests since they are unfortunately not well suited to answering the question: “is this model better than the baseline in general?” For MT, since the pattern was different, we thought it was important to say “how likely is this pattern due to random chance?”
>
> -Complain about the analysis section
> The experiments demonstrate the ordering of the models (tree constrained RecRNs + unsupervised trees > tree constrained RNs > unconstrained RNs) reliable across the tasks. The analysis was just an effort to demonstrate some of the patterns that were being learned.
>
> -Use MNLI instead of SNLI
> We agree that MNLI is a good task to test models on. However, since we are not targeting on solving textual entailment but to investigate whether our model is effective for representation in general, SNLI would be good enough. Again, we have tested our model across 5 different datasets, which should be enough to prove that our model is not overfitting on one particular dataset.

---

### Author Response · Authors · 2018-11-26
**Author response**

We thank the reviewers for their time and the helpful reviews. We first respond to the reviewers’ concern about the novelty of the work. We then answer each reviewer's questions by replying them individually.

The idea of viewing the semantics of a sentence as a conjunction of relations has been quite influential in a number of different semantic formalisms (e.g., Copestake et al., 2005; Smolensky, 1990, as well as more broadly in neo-Davidsonian formalisms), but it has yet to be instantiated directly in a model of sentence representation. Furthermore, the recent successes of relation nets (Santoro et al., 2017) on language-relevant tasks like CLEVR and NLVR, as well as their formal similarity to Smolensky’s (1990) “superpositional representation of conjunction”, suggested this model as a potentially important thing to be. While the simplicity of RNs is a major argument of the original RNs paper, here we instead want to *study a relation-based sentence representation*, and also to *understand* how it needs to be enhanced to perform well. However, a direct application of RNs fails (an interesting result in light of theoretical reasons RNs seem like a good fit), and then we address this with our two extensions, which do in fact make our final models competitive with the best existing models.

To emphasize, the novelties of the paper are:
1) investigation of the relation-based sentence representation.
2) extensions including supervised/unsupervised tree constraints on RNs and Recurrent RNs. The matrix tree theorem has been used in the paper by Koo et al. (have cited relevant papers), and here we adapt this algorithm in our RN formulation.

---

### Meta-Review · Area_Chair1 · 2018-12-14
**Relatively incremental ideas with inconclusive empirical results**

**Confidence:** 5
**Recommendation:** Reject

**Metareview:**

This paper presents two extensions of Relation Networks (RNs) to represent a sentence as a set of relations between words: (1) dependency-based constraints to control the influence of different relations within a sentence and (2) recurrent extension of RNs to propagate information through the tree structure of relations.

Pros:
The notion of relation networks for sentence representation is potentially interesting.

Cons:
The significance of the proposed methods compared to existing variants of TreeRNNs is not clear (R1). R1 requested empirical comparisons against TreeRNNs (since the proposed methods are also of tree shape), but the authors argued back that such experiments are necessary beyond BiLSTM baselines.

Verdict:
Reject. The proposed methods build on relatively incremental ideas and the empirical results are rather inconclusive.